# The Effect of a Histone Deacetylase Inhibitor (AR-42) and Zoledronic Acid on Adult T-Cell Leukemia/Lymphoma Osteolytic Bone Tumors

**DOI:** 10.3390/cancers13205066

**Published:** 2021-10-10

**Authors:** Said M. Elshafae, Nicole A. Kohart, Justin T. Breitbach, Blake E. Hildreth, Thomas J. Rosol

**Affiliations:** 1Department of Veterinary Biosciences, College of Veterinary Medicine, The Ohio State University, Columbus, OH 43210, USA; said.alshafey@fvtm.bu.edu.eg (S.M.E.); kohart.6@buckeyemail.osu.edu (N.A.K.); breitbach.1@buckeyemail.osu.edu (J.T.B.); 2Department of Pathology, Faculty of Veterinary Medicine, Benha University, Kalyubia 13736, Egypt; 3Department of Pathology, School of Medicine, University of Alabama at Birmingham, Birmingham, AL 35294, USA; bhildreth@uabmc.edu; 4Department of Biomedical Sciences, Heritage College of Osteopathic Medicine, Ohio University, Athens, OH 45701, USA

**Keywords:** adult T-cell leukemia/lymphoma, bone, zoledronic acid, AR-42, osteolysis

## Abstract

**Simple Summary:**

Adult T-cell leukemia (ATL) Leukemia is an aggressive, peripheral blood (T-cell) neoplasm associated with human T-cell leukemia virus type 1 (HTLV-1) infection. Recent studies have implicated dysregulated histone deacetylases in ATL pathogenesis. ATL modulates the bone microenvironment of patients and activates osteoclasts (bone resorbing cells) that cause severe bone loss. The objective of this study was to assess the individual and dual effects of AR-42 (HDACi) and zoledronic acid (Zol) on the growth of ATL cells in vitro and in vivo. AR-42 and Zol reduced the viability of ATL cells in vitro. Additionally, Zol and Zol/AR-42 decreased ATL tumor growth and halted osteolysis in bone tumor xenografts in immunodeficient mice in vivo. Our study suggests that dual targeting of ATL cells (using HDACi) and bone osteoclasts (using bisphosphonates) may be exploited as a valuable approach to reduce bone tumor burden and improve the life quality of ATL patients.

**Abstract:**

Adult T-cell leukemia/lymphoma (ATL) is an intractable disease affecting nearly 4% of Human T-cell Leukemia Virus Type 1 (HTLV-1) carriers. Acute ATL has a unique interaction with bone characterized by aggressive bone invasion, osteolytic metastasis, and hypercalcemia. We hypothesized that dual tumor and bone-targeted therapies would decrease tumor burden in bone, the incidence of metastasis, and ATL-associated osteolysis. Our goal was to evaluate dual targeting of both ATL bone tumors and the bone microenvironment using an anti-tumor HDACi (AR-42) and an osteoclast inhibitor (zoledronic acid, Zol), alone and in combination. Our results showed that AR-42, Zol, and AR-42/Zol significantly decreased the viability of multiple ATL cancer cell lines in vitro. Zol and AR-42/Zol decreased tumor growth in vivo. Zol ± AR-42 significantly decreased ATL-associated bone resorption and promoted new bone formation. AR-42-treated ATL cells had increased mRNA levels of *PTHrP*, *ENPP2* (*autotaxin*) and *MIP-1*α, and *TAX* viral gene expression. AR-42 alone had no significant effect on tumor growth or osteolysis in mice. These findings indicate that Zol adjuvant therapy has the potential to reduce growth of ATL in bone and its associated osteolysis.

## 1. Introduction

Nearly 4% of Human T-cell Leukemia Virus Type 1 (HTLV-1) infected patients will develop Adult T-cell Leukemia/Lymphoma (ATL) after a 20–40-year latency period. ATL is an aggressive and fatal hematologic malignancy associated with a poor survival rate of less than a year due to its rapid progression, overwhelming tumor burden, immunosuppression, and resistance to traditional chemotherapies [1,2]. Patients with ATL frequently have generalized lymphadenopathy, skin lesions, severe osteolytic bone metastases, and paraneoplastic hypercalcemia [3,4,5,6]. The clinicopathologic features of ATL in bone indicate a need to investigate osteolytic bone metastasis and establish some novel dual tumor-bone therapies.

The HTLV-1 virus expresses two oncogenes, Tax and Hbz, that promote ATL tumorigenesis and directly affect bone cells. Tax is a transcriptional activator that activates several bone-related factors including, parathyroid hormone-related protein (PTHrP), transforming growth factor-β (TGF-β), tumor necrosis factor-α (TNF-α), and macrophage inflammatory protein-1α (MIP-1α) [7,8]. Tax also inhibits tumor-suppressor gene (p53) function and induces tumor formation through constitutive activation of the alternative NF-kB pathway [2,9]. Hbz promotes lymphocyte transformation, tumor proliferation, viral persistence in vitro and in vivo, and disrupts WNT signaling [10,11,12,13]. HBZ upregulation was found to be implicated in the pathogenesis of bone loss in in adult T cell leukemia/lymphoma (ATL) through regulation of RANKL expression [14]. Therefore, Tax and Hbz promote tumor cell survival and disrupt normal bone remodeling by activation of resident bone-resorbing osteoclasts and inhibition of bone-forming osteoblasts. When osteoclasts are activated, they function to resorb preexisting bone. The resorbed bone releases factors normally stored in the mineralized matrix, including growth factors and minerals that promote tumor growth, such as TGF-β, insulin-like growth factors (IGF), and bone morphogenetic proteins (BMP) [15]. The process of pathologic bone resorption results in a favorable tumor microenvironment for cancer cell growth, metastasis, and drug resistance [6,16]. This cascade of events is known as the ‘vicious cycle’ [17]. The factors induced by Tax and Hbz also have a tumor-promoting effect by modulating the bone microenvironment. Targeting the bone microenvironment and the key drivers of vicious cycle regulated by Tax and Hbz would improve traditional therapeutic regimens.

A promising class of anti-cancer drugs are histone deacetylase inhibitors (HDACi). They have been shown to inhibit tumor growth, induce differentiation of cancer cells, and promote tumor cell apoptosis. HDACs are a family of cellular enzymes that remove acetyl groups from histone proteins thus repressing transcription and resulting in epigenetic silencing. These epigenetic effects on DNA regulation can promote carcinogenesis by silencing tumor suppressors, such as P53. HDACi counteract pro-tumorigenic effects by modulating histones to prevent acetylation and silencing of transcription, thereby allowing tumor suppressors genes to be transcribed [18]. HDACi target several cytokine signaling pathways and induce apoptosis by regulating genes in the intrinsic and extrinsic apoptotic pathways [19]. The pathogenesis of ATL has been linked to dysregulated NF-kB signaling, which is also a downstream effector of HDACi [9]. Constitutive expression of NF-kB in ATL cells has been implicated in its resistance to cancer therapy and anti-apoptotic function [20]. It also has been shown to promote humoral hypercalcemia of malignancy (HHM) and tumor-associated osteolysis through activation of PTHrP transcription [20,21,22]. Furthermore, the long latency of HTLV-1-infected cells before the emergence of ATL suggests several genetic and/or epigenetic changes are responsible for induction of disease. The epigenetic changes in histones preceding ATL indicate there is potential merit for the use of HDACi as a treatment for ATL [23]. However, the role of HDACi on osteolytic bone metastasis and the ATL tumor-bone interactions remain unknown.

Patients with ATL frequently have high serum concentrations of PTHrP, IL-1, TGF-β, MIP-1α, and IL-6 that act locally and at distant sites to promote bone resorbing osteoclasts. The tumor-induced, pathologic osteoclastic activity results in diffuse or local skeletal resorption that contributes to significant patient morbidity. Bisphosphonates are potent anti-resorptive therapeutic agents that are used in the treatment of conditions associated with excessive bone resorption such as osteoporosis, Paget’s disease, and tumor-induced bone lysis [24]. Bisphosphonates inhibit the mevalonate pathway and associated Ras signaling in activated osteoclasts. Disruption of Ras signaling inhibits osteoclast development and prevents the formation of the ruffled border necessary for osteoclasts to resorb bone. Zoledronic acid (Zol) is a third-generation bisphosphonate that has been shown to have direct anti-tumor and cytotoxic effects in breast and prostate cancer, myeloma, chronic myeloid leukemia, and ATL [20,25,26]. Zol inhibits the prenylation of Ras and Ras-related proteins that decreases leukemia cell proliferation and induces cell cycle arrest at S-phase and apoptosis [25]. The dual effects of Zol on bone and ATL cells have been demonstrated in Tax transgenic mice (Tax+), where Zol prevented osteolytic bone lysis and decreased tumor burden in Tax+ mice [27].

There are several animal models of ATL, including primates, rabbits, rats, and humanized, transgenic, and xenograft mouse models. These models were developed to investigate HTLV-1/ATL transformation and proliferation, the roles of oncoproteins, or for testing of therapeutic agents for tumor growth and/or paraneoplastic syndromes [28]. However, there is a lack of well-defined animal models for investigations of tumor-bone interactions in ATL. We successfully developed a mouse model of intraosseous ATL using MT-2 cells that express viral proteins, Tax and Hbz, and pro-osteolytic genes typically found in ATL cells. MT-2 cells are an in vitro transformed, patient-derived ATL cell line that grows in immunosuppressed mice and results in aggressive osteolytic lesions when injected into the tibia, similar to those observed in ATL patients [6,29]. We transduced MT-2 cells with a lentivirus containing the luciferase gene so that noninvasive tumor engraftment, metastasis, and growth could be measured in vivo. 

The goals of this study were to evaluate the independent and potential synergistic role of combined anti-tumor HDACi, AR-42, and bisphosphonate, (Zol) therapy on ATL bone metastasis and ATL-associated osteolytic lesions.

## 2. Materials and Methods

### 2.1. Cells and Reagents

MT-2 cells were generated by coculturing cord lymphocytes with leukemia cells derived from a patient with ATL [30]. Primary cells were superinfected with HTLV-1 to induce a high viral load, high Tax expression, and the ability to grow in vitro [31]. The HT1RV cell line was established by culturing of RVATL cells with SLB-I cells (HTLV1 positive cell line that had been lethally irradiated) in Iscove culture medium [32]. The HT1RV cell line is superinfected with HTLV-1 and has high Tax expression, while the ATLED cell line was derived from an ATL patient and does not express Tax [33,34]. Cells were maintained in 1640 RPMI medium (Corning, Manassas, VA, USA) supplemented with 10% FBS, 2 mM glutamine, 100 U/mL penicillin, and 100 µg/mL streptomycin at 37 °C and 5% carbon dioxide. AR-42 was obtained from ARNO therapeutics (Parsippany, NJ, USA). Zoledronic acid (Zol) was purchased from The Ohio State University James Cancer Center In-clinic pharmacy (Columbus, OH, USA).

### 2.2. Transduction of MT-2 Cells with Luciferase

To monitor tumor cell engraftment, growth, progression, and metastasis in vivo, MT-2 cells were transduced with a lentiviral vector containing the luciferase enzyme. 5 × 10^5^ MT-2 cells were pelleted in a 1.5 mL Eppendorf tube and resuspended in 25 μL of freshly concentrated lentiviral vector (The pCDH-LTR-1-luc-EF1α-copGFP vector). 125 μL of complete RPMI medium was added to the vector and cells. The total volume (150 μL) was added to a single well of a 96-well plate. 8 ug/mL of polybrene was added to the well containing the vector suspension. The plate was wrapped in parafilm and spun at 2000× *g* for 2 h at room temperature. The parafilm was removed and the plate containing the transduced cells was incubated for 1 h at 37 °C and 5% CO_2_. The pellet was resuspended in 1 mL of medium and transferred to a 6-well culture plate containing 1 mL of complete culture medium. After 48 h, the transduced cells were transferred to a 25 cm^2^ flask containing 10 mL of complete RPMI medium and allowed to grow.

### 2.3. Bioluminescent Imaging

MT-2-Luc (MT-2 luciferase transduced cells) tumor growth and metastasis in mice were monitored using bioluminescent imaging. Initial pilot studies were performed to confirm that bioluminescent signals were representative of tumor growth within the tibia and for the detection of metastasis. Mice were imaged immediately following injection to confirm the intratibial inoculation and then once a week for the duration of the study. Bioluminescent imaging was performed using the in vivo imaging system IVIS 100 (Caliper Life Sciences, Hopkinton, MA, USA) as previously described [35]. Briefly, 0.15 mL of sterile DPBS containing 4.5 mg of D-luciferin (Caliper Life Sciences) was injected intraperitoneally and imaging started 5 min later. Serial images were taken every 2 min until peak photon emission was obtained at approximately 10 min. Photon signal intensity of each tibia was quantified using Living Image software version 2.50 (Caliper Life Sciences). Tumor growth was determined based on the fold change in total flux (photons/second) from the day of injection till the day of termination.

### 2.4. Cell Viability Assays

To measure the cell viability following treatment with AR-42, Zol, or AR-42/Zol, MTT assays were performed. MT-2, ATLED, and HT1RV cells were plated at a density of 50,000 cells/well and co-cultured with various concentrations of either AR-42 (0–3 μM), Zol (0–250 μM), AR-42/Zol, or vehicle for up to 3 days in 96-well plates (Corning, Manassas, VA, USA). Cell viability was determined by measuring the mitochondrial-dependent conversion of the 3-(4,5-dimethylithiaZol-2-yl)-2,5-diphenyl tetrazolium salt (MTT) (Biomatik, Cat. #A3338, Wilmington, DE, USA) to a colored formazan product. At 72 h, 20 µM of 5 mg/mL MTT was added to each well and cells were incubated for 4 h. After incubation, 100 µL of DMSO was added, and the absorbance was measured using an ELISA plate reader at a wavelength of 520 nm.

### 2.5. Treatment Study

Five to seven-week-old female NOD-scid IL2Rgammanull (NSG) mice (Target Validated Shared Resource, The Ohio State University, Columbus, OH, USA) were housed and treated in accordance with the ULAR guidelines. Experimental protocols were approved by the Institutional Animal Care and Use Committee. Intratibial injections were performed to recapitulate bone metastasis. Mice were anesthetized in an induction chamber with 3% isoflurane and maintained with 2.5% isoflurane. The right rear limb was shaved with a disposable razor to remove the fur. The exposed tibia was scrubbed with 70% alcohol-soaked gauze. The leg was held so the knee joint was at a 90° angle and a 27-ga, 0.5-inch needle was introduced through the patellar ligament and into the tibial marrow space through the articular cartilage. A Hamilton syringe was used to inject 50,000 MT-2 cells in 10 μL RPMI 1640 medium into the marrow cavity of the right tibia.

Treatment was started three days following inoculation. Mice were randomly assigned into four groups: Vehicle control, AR-42 (20 mg/kg 3 times per week, oral gavage), Zol (0.1 mg/kg twice per week, subcutaneously), or AR-42/Zol for 3 weeks. Mice were monitored by checking their body weight and overall physical presentation three times a week, weighed twice a week, and bioluminescent imaged once a week for the duration of the study.

### 2.6. Histopathology

A complete gross necropsy was performed following euthanasia. Tibias were excised, defleshed, and fixed in 10% neutral-buffered formalin for 48–72 h. After fixation, tibias were placed in 70% ethanol for 24 h to prepare for radiographic imaging. Following imaging, tibias were decalcified in 10% EDTA (pH 7.4) for two weeks at 4 °C and embedded in paraffin. Tibial sections were stained with H&E for histopathologic evaluation.

### 2.7. Radiographic Imaging

Tibias and femurs were placed centrally on a Faxitron laboratory radiography system LX-60 (Faxitron X-ray Corp., Wheeling, IL, USA) imaging platform. Each leg was exposed to 30 KVP for 6 s. High resolution radiographs were used to quantitate bone loss for each image using Bioquant software (Bioquant Osteo 2013 Version 13.2.60; Bioquant Image Analysis Corporation, Nashville, TN, USA). DICOM images were uploaded, the perimeter of each tibia outlined, and the total tibial area measured. Areas of radiolucency were then outlined and measured. Percent of tumor-associated bone loss (osteolysis) was calculated by dividing the area of radiolucency (bone loss) by the total tibial area.

### 2.8. Microcomputed Tomography (μCT)

MT2 inoculated tibias were placed in a holder (17-mm) and images were taken by micro-computed tomography (μCT) (μCT-40; Scanco Medical). Inveon Research Workplace 3.0 software was used to analyze μCT DICOM images (Siemens Medical Solutions Inc., Knoxville, TN, USA). 3D reconstructions of the bone were produced [36].

### 2.9. Real-Time Reverse Transcription-PCR (qRT-PCR)

MT-2 cells were seeded in 6-well culture plates at a density of 200,000 cells per well. Medium (2 mls) containing either 1% DMSO, 0.1 or 2.5 µM of AR-42, 10 or 50 µM of Zol, or the combinations of AR-42/Zol (low and high doses) were added to each well in triplicate. After 24 h incubation, cells were collected and RNA was extracted using the QuickGene DNA cultured cell HC kit (AutoGen, Holliston, MA, USA). Reverse transcription (RT) and quantitative real-time polymerase chain reaction (qRT-PCR) were performed as previously described [37,38]. Specific human oligonucleotide primers were used for *GAPDH*, *PTHrP*, *MIP-1α*, *ENPP2*, *Wnt5a*, *IL-6*, *CXCR4*, *TAX* and *HBZ* (Table 1). Relative gene expression was normalized to GAPDH. All RT-PCR reactions were performed in triplicate.

### 2.10. Statistical Analysis

Data were analyzed using Graph Pad Prism 6.0 software (San Diego, CA, USA). Quantitative RT-PCR values were normalized to GAPDH mRNA and were expressed as the fold difference between the groups (mean ± SD). A *p* ≤ 0.05 was considered to be statistically significant. Outcome variables were compared between groups with a parametric one-way ANOVA or nonparametric Kruskal-Wallis test. Post-hoc comparisons between treatment groups and the control group were performed using Dunnett’s multiple comparisons test to adjust for multiplicity.

## 3. Results

### 3.1. MTT Viability Assays

Treatment with AR-42 significantly (*p* < 0.01) reduced cell viability in MT-2, ATLED, and HT1RV cell lines, which was independent of Tax gene expression. AR-42 significantly reduced cell viability at concentrations of 0.1–3 µM for the MT-2, HT1RV and ATLED cell lines at 72 h (Figure 1A). Zol significantly reduced cell viability in the MT-2 and HT1RV cell lines at 50–250 µM (*p* < 0.0001) and ATLED at 100–250 µM at 72 h of treatment (*p* < 0.01) (Figure 1B). A trend for greater reduction in cell viability was observed in cells treated with the combination of AR-42 and Zol compared to either agent alone, and was most notable in the HT1RV and ATLED cell lines. There was a marked decline in the viability of MT-2, HT1RV, and ATLED from 0.5–2.5 µM of AR-42/50–250 µM Zol (*p* < 0.001) (Figure 1C).

### 3.2. Development of MT-2 Intratibial Mouse Model

We successfully developed a model of ATL in bone with associated pathologic lesions, similar to those seen in HTLV-1-infected and ATL patients. MT-2 cells were transduced with the luciferase gene to allow for non-invasive imaging. Localized bioluminescent signal was seen in the tibias of NSG mice immediately post-injection, and after 4 weeks mice developed osteolytic lesions. At 4-weeks following injection, μCT images showed that tumor-bearing NSG mice had marked multifocal to coalescing areas of pathologic bone loss throughout the proximal metaphysis and spanning to mid-diaphysis, which resulted in pathologic fracture (Figure 2). The lytic bone lesions resembled those of ATL patients presenting with bone metastasis. The rapid tumor development, ability to monitor tumor cell growth noninvasively in vivo, and the pathologic phenotype similar to what is seen in clinical patients make the MT-2 intratibial model the ideal model to test novel therapeutics and factors involved in the ATL-bone communication.

### 3.3. Bioluminescence, Tumor Growth and Tibial Tumor Burden

Viable intratibial tumor growth was measured and compared between groups using bioluminescence weekly and on the last day of the study. Mice treated with Zol and AR-42/Zol had a significant decrease in tumor growth rate compared to control mice starting from week 2 post intratibial injection (*p* ≤ 0.05) (Figure 3A,B). On the day of sacrifice, total tibial tumor burden was also significantly decreased in Zol and AR-42/Zol-treated mice (*p* < 0.01) (Figure 3C). Unexpectedly, the mice treated with AR-42 alone had a relative increase in both the rate of tumor growth and tibial tumor burden on the day of sacrifice compared to controls, Zol, and AR-42/Zol-treated groups, but these findings were not statistically significant (Figure 3B,C).

### 3.4. Faxitron Radiography and Analysis

Marked focal to multifocal to coalescing areas of radiolucency (osteolysis) spanning from the proximal tibia throughout the diaphysis in the control and AR-42-treated mice were seen (Figure 4A,B). Zol and AR-42/Zol-treated groups showed significantly less radiographic osteolysis characterized by smooth, intact cortices and small focal radiolucent areas (Figure 4C,D). Zol significantly decreased the area of radiographic pathologic bone loss compared to the control group (*p* < 0.05) (Figure 4E); however, AR-42-treated mice did not show significant changes compared to controls (Figure 4E).

### 3.5. Histopathology of Intratibial Tumors

Histopathologic examination of the MT-2-luc tibial xenografts in control and in AR-42-treated mice revealed dense sheets of large atypical round cells with indistinct cellular borders spanning from the proximal metaphysis throughout the tibial diaphysis, frequently invading through eroded cortices and into surrounding soft tissues. Tumor-bearing tibias of control and AR-42-treated mice had marked pathologic osteolysis characterized by frequent eroded surfaces lined by active multi-nucleated osteoclasts (Figure 5A–F). Necrosis was present throughout the marrow cavity in AR-42-treated mice. In contrast to control and AR-42-treated mice, there was a significant reduction in eroded surfaces and osteolysis in the Zol and AR-42/Zol-treated mice. Cortical boundaries remained intact and marked intramedullary and periosteal new bone formation was evident, lined by increased numbers of plump, active osteoblasts in the Zol and AR-42/Zol-treated mice (Figure 5G–L). Necrosis was present from the proximal tibia and throughout the medullary cavity.

### 3.6. qRT-PCR

Treatment of MT-2 cells with AR-42 at 0.1 µM and 2.5 µM significantly increased the mRNA expression of *ENPP2*, *MIP-1α* and *TAX* (*p* < 0.05) (Figure 6A,B). AR-42 (0.1 µM) increased *IL-6* and *PTHrP* mRNA expression in MT-2 (*p* < 0.01) with a trend for *CXCR4* mRNA levels to be decreased at 2.5 μM (*p* = 0.07) (Figure 6A,B). Treatment with Zol did not induce significant changes in the mRNA expression of the tumor-derived osteolytic factors at any concentration (Figure 6C,D). The combination of AR-42 and Zol resulted in a similar profile as AR-42 treatment alone. Osteolytic factors, including *ENPP2* and *MIP-1*α mRNA were increased in MT2 cells treated with 0.1 µM AR-42 and 10 µM Zol, and with 0.5 µM AR-42 and 20 µM Zol (*p* < 0.05) while there was marked elevation in the levels of *PTHrP* and *IL-6* mRNA in AR-42/Zol-treated MT-2 cells at 0.1 µM AR-42 and 10 µM Zol (*p* < 0.05) (Figure 6E,F). No significant changes were observed in *CXCR4* or *WNT5a* mRNA. AR-42/Zol increased *TAX* gene expression (*p* < 0.01) at doses of 0.1/10 µM and 0.5/20 µM of AR-42/Zol combination (Figure 6E,F).

## 4. Discussion

Aggressive bone invasion, osteolytic metastasis, and hypercalcemia of malignancy occur in nearly 80% of patients with acute ATL [15,39]. Circulating tumor-derived PTHrP, MIP-1α, IL-1, IL-6, and TNF-α have been identified in the serum of ATL patients and are factors underlying the clinical tumor-associated bone manifestations [4,6,21,40]. Osteolytic lesions in patients with ATL occur in various bones including the skull, pelvis, spine, and long bones [41,42]. Bone metastasis and associated skeletal complications lead to pathologic fractures, nerve compression, pain, and dramatically worsen patients’ quality of life and prognosis. Similar manifestations in bone occur in patients with metastatic breast, prostate, and lung cancers [43]. To investigate the mechanism of osteolysis in bone lymphoma and its response to therapy, we established a novel preclinical animal model that recapitulates the aggressive osteolytic lesions observed in ATL patients. We evaluated the effects of an innovative therapeutic combination of an HDACi, AR-42 with a bisphosphonate, Zoledronic acid (Zol) on ATL osteolytic bone metastasis.

Patients with ATL have a poor prognosis due to the rapidly aggressive nature of the disease and the intrinsic resistance of tumor cells to traditional chemotherapy. Several sole and combination therapeutic approaches have been used in ATL with limited success. These include first and second generation polychemotherapies including cyclophosphamide, hydroxydaunorubicin, vincristine, prednisone, and CHOP-like regimens; nucleoside analogues; topoisomerase inhibitors; antiviral therapy zidovudine and interferon-γ; arsenic trioxide; monoclonal antibodies, and autologous hematopoietic stem cell transplants [44,45]. In recent years, various targeted therapies using inhibitors and small molecules have been investigated to assess the potential of their use in clinical trials. These include proteasome inhibitors, NF-kB inhibitors, reverse transcriptase inhibitors, and small molecule inhibitors of Bcl-2, Bcl-X(L), and Bcl-w [46]. 

Preclinical models of ATL have been instrumental to investigate the efficacy of traditional and novel therapies in vivo, particularly xenograft, transgenic, and the “humanized” mouse models. However, while they have been proven useful, a single model is not able to faithfully recapitulate all aspects of the disease. In our study, the MT-2 intratibial model consistently resulted in tumor growth and distant metastasis, similar to the aggressive osteolytic lesions observed in ATL patients. Intratibial injections were performed to recapitulate the direct tumor-bone interaction in ATL bone metastasis. Metastatic ATL cells interact with local osteoblasts, osteoclasts, stromal cells, and hematopoietic constituents. Because tumor progression, burden, and metastasis are challenging to be assessed in vivo following tibial injections, MT-2 cells were transduced with a lentivirus containing the luciferase gene. This allowed for a noninvasive, sensitive, and accurate monitoring of tumor growth and metastasis and in vivo response to therapy in our study. 

HDACi have several cellular functions and antitumor capabilities through transcriptional reactivation of dormant tumor suppressor genes and regulation of genes that are critical to cell proliferation, cell cycle, apoptosis, cytoskeletal modifications and angiogenesis [47]. Tax induces p53 inactivation and interferes with p53 function in ATL cells [48]. Tax also has trans-activating effects on multiple signaling pathways including CREB and NF-kB that result in differential effects on cellular proliferation and transformation. Downstream cytokines regulated by NF-kB and CREB signaling genes have a critical role in osteoclast and osteoblast function, which may in part be responsible for the pathologic bone modeling in ATL patients [7,49]. The overlapping targets of HDACi with Tax on the regulation of gene transcription suggest their use as potential therapeutic agents in ATL.

HDACi have been shown to inhibit NF-kB/DNA binding in HTLV-1 infected T-cells thus increasing the level of the inhibitory subunit of NF-kB (IkB) and downregulating NF-kB entry into the nucleus [50]. Recent studies showed that NF-kB complexes modulates the PTHrP P2 promoter [37]. PTHrP plays a critical role in osteoclastic bone resorption and humoral hypercalcemia of malignancy (HHM) in ATL patients.

We evaluated the effect of AR-42, a unique HDACi, on ATL bone metastasis and on osteolytic lesions. There was a significant in vitro effect on ATL cell viability independent of Tax expression. A previous study also reported a significant reduction in cell viability in MT-2 cells at 48-h with similar AR-42 concentrations. This study reported that AR-42-induced apoptosis and a dose-dependent hyperacetylation of histone H3 and an increase in cytochrome C and cleaved poly (ADP-ribose) polymerase, both indicators of apoptosis [51].

Despite the significant antineoplastic effect seen in vitro, there was no effect of AR-42 on tumor progression or bone lesions in vivo. Several reasons may account for these findings. Typical oral gavage dosing schedules of AR-42 range from 75 mg/kg three times a week to 25 mg/kg every day [18,52,53]. These regimens had been tested in SCID and nude mice; however, there was limited information on dosing of AR-42 in NSG mice. We selected a 20 mg/kg three times a week dosing schedule to closely emulate the dose and therapeutic regimen in human patients. Work in our lab from previous studies had found doses of 25 mg/kg every day in NSG mice resulted in death within 1–2 weeks following the start of therapy. Use of a dietary formulation of AR-42 has shown effective as anti-tumor therapy and would also be worth investigating in this model [51]. Further work is needed to optimize the oral dosage and frequency of administration of AR-42.

Bone metastasis is a frequent complication in breast and prostate cancer and results in significant patient morbidity and reduction in quality of life. The HDACi, vorinostat, decreased tumor growth and associated osteolysis in mice with breast or prostate cancer cells in their tibias; however, diffuse osteopenia was also observed independent of tumor cell activity [54]. Vorinostat increased the expression of tumor-derived factors known to promote osteolysis, such as PTHrP, and this was the proposed mechanism for the diffuse bone loss associated with treatment. MT-2-treated cells with AR-42 in our study had increased mRNA expression of several pro-osteolytic factors, including PTHrP. Messenger RNA for HTLV-1 viral oncogene, Tax, was also increased in treated cells. This finding was consistent with other studies that showed an increase in Tax expression following treatment with some HDACi, such as Valproate, Vorinostat and Romidepsin [55,56,57,58,59]. Previous reports have found a role for HDACi in HIV-reactivation in patients with latent HIV infection [60,61]. When CD4+ T-cell lines with HIV-1 latency were treated with AR-42, histone acetylation resulted in HIV reactivation [62]. A similar mechanism of viral activation could be hypothesized to explain the increase in Tax expression following treatment of MT-2 cells. Tax has been shown to increase gene expression of osteolytic factors, such as PTHrP. AR-42 treatment and increased mRNA expression of Tax could suggest a potential mechanism for a downstream increase in PTHrP mRNA in the AR-42-treated cells. Future studies to investigate the relationship of HDACi and Tax-induced osteolytic factors would be of value to further understand the mechanisms of ATL-induced osteolysis.

Bisphosphonates are potent inhibitors of bone resorption and are approved for patients with malignant bone disease to prevent skeletal-related events. Zoledronic acid is a third generation, nitrogen-containing bisphosphonate that binds to bone, is taken up by active osteoclasts, and induces osteoclast apoptosis [63]. Zol works to inhibit the mevalonate pathway and reduces activity of farnesyl pyrophosphate synthase, resulting in a decrease of farnesyl pyrophosphate and geranylgeranyl pyrophosphate necessary for the activity of cellular proteins, Ras, Rho, Rac, and Rab [64,65,66]. In addition to the inhibition of osteoclasts, high-dose Zol has direct anti-tumor effects. Induction of tumor cell apoptosis, decrease of matrix invasion and adhesion to bone, inhibition of angiogenesis, and stimulation of immune surveillance have all been attributed to Zol therapy in several cancer types, which suggest their potential use in the metastatic cascade of ATL [67].

Zol-treated Tax+ transgenic mice had a significant decrease in soft tissue tumor development and tumor-associated bone destruction, and an increase in survival [27]. Decreased cell viability and inhibition of protein prenylation was observed in Tax+ cell lines when treated with Zol concentrations as low as 10 µM [68]. When Zol therapy was evaluated in a xenograft model of ATL, tumor burden, osteoclast activity, and hypercalcemia were decreased [20]. Here we investigated the effects of Zol on established ATL bone metastasis and associated osteolytic lesions. Zol significantly reduced tumor growth, distant metastasis, and tumor-associated osteolysis. The anti-tumor effect in vivo was possibly a combined direct and indirect anti-tumor effect. Evidence to support contribution from a direct anti-cancer cell effect was supported by the decrease in cell viability in vitro in the three ATL cell lines treated with varying concentrations of Zol. Furthermore, the effect of Zol on cell viability was independent of Tax gene expression. 

An anti-cancer effect of Zol through modulation of the bone microenvironment also played a role in vivo in our study. We have shown radiographical evidence of osteosclerosis in the proximal and distal epiphyses of the tibiae, away from the site of tumor injection and growth, in Zol and AR-42/Zol-treated mice. In addition, there was a reduction in osteoclasts histologically in these treatment groups at the tumor/bone interface. These cumulative findings indicated that Zol decreased osteoclast activity. Zol-mediated osteoclast inhibition could prevent tumor-induced remodeling of the bone microenvironment to prevent tumor growth and deprive the ATL cells of bone-derived growth factors. The findings from this experiment emphasize the significant role of the bone microenvironment in tumor progression. It has been speculated that ATL cells grow and respond favorably to the constituents released from bone due to tumor-induced osteoclast activity. Future studies with this model to investigate and/or inhibit the individual effects of bone-derived growth factors on ATL cells would be important to further elucidate the molecular pathogenesis of the ATL tumor-bone communication and to identify additional therapeutic targets.

Many research approaches have been employed to extend the survival of ATL patients. These have been directed against the tumor and virus, including combinations of chemotherapeutics, targeted therapies, and antiviral agents [69]. However, the combined approach for both tumor and microenvironment remains largely unexplored in ATL and ATL bone metastasis. When prostate cancer cells were treated with HDACi, suberoylanilide hydroxamic acid (SAHA) and Zol in vitro, an increase in cell death was due to a synergistic anti-tumor effect of both SAHA and Zol inhibition of geranylgeranylation resulting in increased cell death [70]. A potent synergistic effect of the HDACi panobinostat with Zol was observed in prostate cancer cell lines in vitro and in vivo in a prostate cancer xenograft model [71]. In the current study, we evaluated the potential synergistic effects of an HDACi, AR-42, with Zol on tumor cell growth and metastasis and the effect on the tumor-bone communication. In vitro, individual treatment with AR-42 or Zol alone resulted in an increase in cell death in ATL cell lines. However, treatment of cells in vitro with the combination therapy resulted in a greater reduction of cell viability when compared to individual therapy, most notably in HT1RV and ATLED cells. These data suggest that the addition of Zol to AR-42 treatment may be beneficial to cells initially resistant to HDACi therapy and could also suggest an increase in potency when the cells are treated with the combination of AR-42 and Zol. The combination may also reduce the need for higher doses of AR-42, which will reduce the risk of toxicity in clinical patients.

## 5. Conclusions

In this study, we report the development of a novel bioluminescent animal model of ATL that successfully and consistently recapitulates the phenotypic manifestations of ATL bone metastasis and HTLV-1-associated osteolytic lesions. We showed that Zol and the combination of Zol with AR-42 was an effective therapy to decrease ATL tumor growth and associated osteolytic lesions through direct and indirect anti-tumor effects. While AR-42 was not effective as a single agent, the addition of AR-42 to Zol did not inhibit the effectiveness of Zol. Of interest, we found an increase in Tax expression following AR-42 treatment, suggesting that further investigation in the effects of HDACi treatment on HTLV-1 viral oncogene expression is merited. Furthermore, this study provides evidence of the importance of the tumor-bone communication in ATL and indicates the need for the development and testing of novel tumor-bone targeted therapies in ATL.

## Figures and Tables

**Figure 1 cancers-13-05066-f001:**
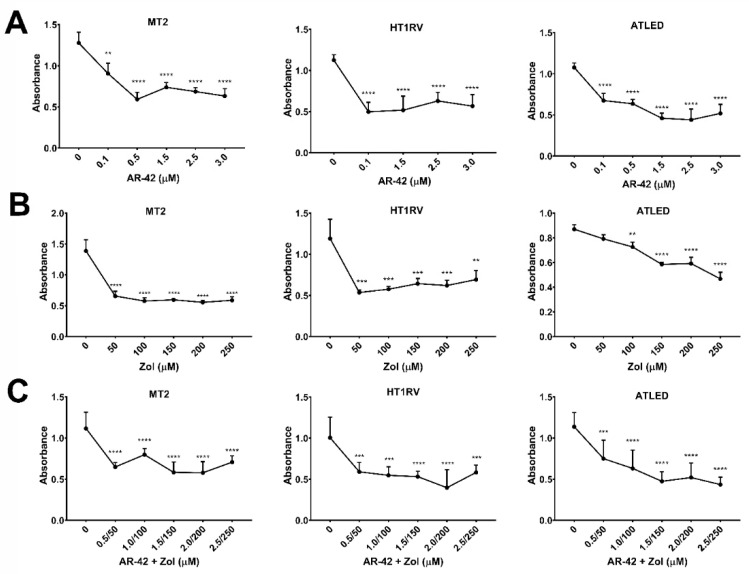
The Effect of AR-42 (**A**), Zol (**B**), and AR-42/Zol (**C**) on the viability of MT2, HT1RV and ATLED cell lines after 72 h using the MTT assay. Data are represented by means ± standard deviation (SD) (*N* = 6–9) for each experiment. Significant differences between control cells (0) and treatments are indicated as ** *p*  <  0.001, *** *p* < 0.0001, **** *p* < 0.00001.

**Figure 2 cancers-13-05066-f002:**
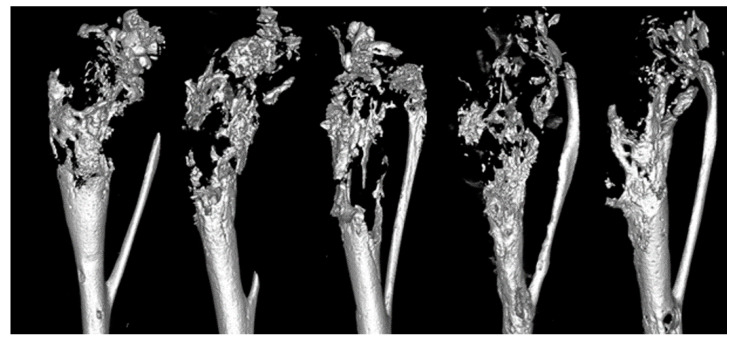
Micro-CT 3D reconstructions of tibias from NSG mice bearing MT-2 cells. MT-2 cell inoculated tibias of NSG mice showed marked osteolysis in the epiphysis and metaphysis.

**Figure 3 cancers-13-05066-f003:**
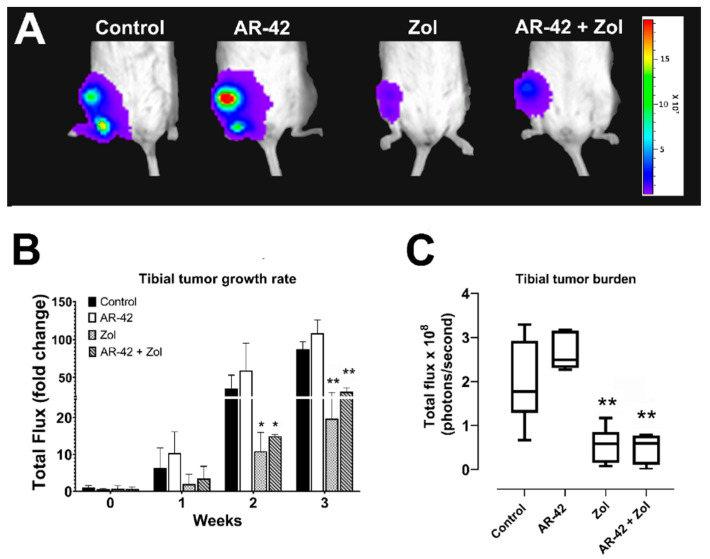
(**A**) Representative bioluminescent images of MT-2-inoculated mice after treatment with AR-42, Zoledronic acid, combination, and vehicle (control) on the day of sacrifice. (**B**) Bar graph of weekly bioluminescence (fold change in photons/sec compared to control) measured in MT2 intratibial xenografts following treatment; data are represented by means ± standard deviation (SD). (**C**) Box and whisker plots of bioluminescence (photons/sec) measured on the day of sacrifice in treated vs. control mice. The median is indicated by the horizontal line inside the box, and the box encompasses the interquartile range (25–75th percentiles) while whiskers span 10–90th percentile of data. Significant differences between different groups and control are indicated as * *p* ≤ 0.05, ** *p* ≤ 0.01.

**Figure 4 cancers-13-05066-f004:**
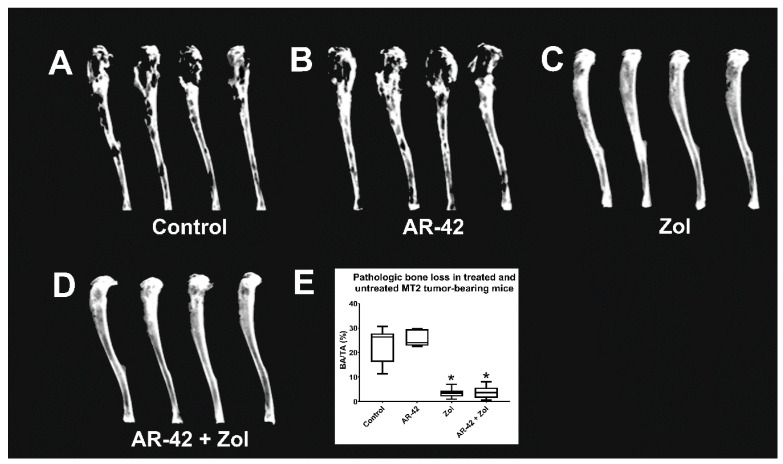
(**A**–**D**) Radiographic images of MT-2-luc xenografted tibias of mice treated with either AR-42, Zoledronic acid, the combination, or vehicle. (**E**) Tumor-associated bone loss in treated and untreated MT-2-luc xenografted mice. BA = area of bone loss; TA = total tibial area; % BA/TA = percentage of area of bone loss per total tibial area. The box encompasses the interquartile range (25–75th percentiles), whereas whiskers span the 10–90th percentile of data. The median is denoted by the line within the box. Significant difference between different groups and control is indicated as * *p* ≤ 0.05.

**Figure 5 cancers-13-05066-f005:**
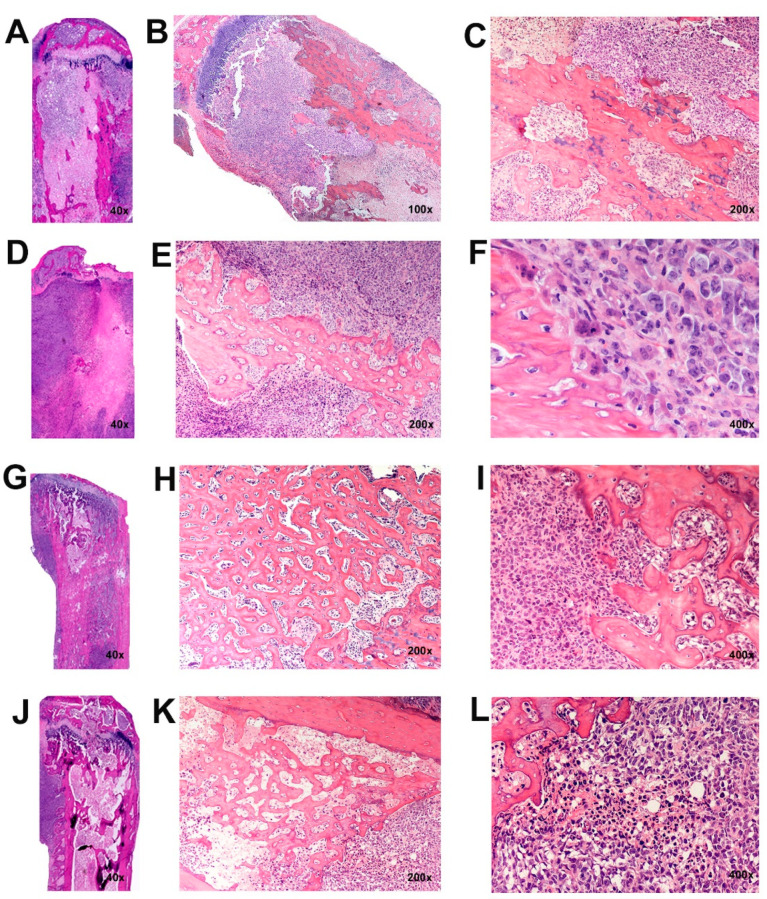
Histopathology of MT-2-luc xenografted tibias in treated and untreated mice. (**A**–**C**) In control mice, marked pathologic osteolysis characterized by eroded surfaces lined by active osteoclasts was seen. Marked necrosis was present throughout the marrow cavity. In AR-42-treated mice (**D**–**F**), bone tumors and osteolysis closely resembled those of the control mice. In Zol-treated mice (**G**–**I**), a marked decrease in eroded surfaces and bone loss with osteoblast activation and formation of intramedullary woven trabecular bone was present. Mice treated with both Zol, and AR-42 (**J**–**L**) had a similar phenotype as the Zol group.

**Figure 6 cancers-13-05066-f006:**
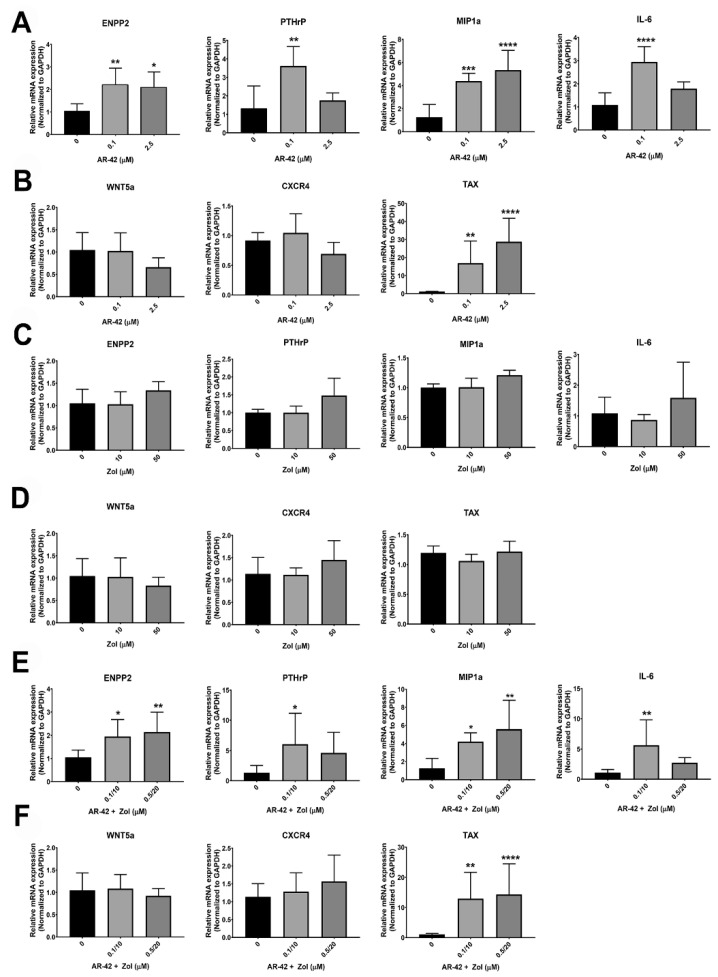
The graphs represent the relative mRNA expression of *ENPP2*, *PTHrP*, *MIP-1α*, *IL-6*, *WNT5a*, *CXCR4* and *TAX* genes in MT2 cells after treatment with either; (**A**,**B**) AR-42 (0, 0.1, 2.5 µM); (**C**,**D**) Zol acid (0, 10, 50 µM); (**E**,**F**) AR-42/Zol acid (0, 0.1/10, 0.5/20 µM in combination). Data are represented by means ± standard deviation (SD) and significant differences between control MT2 (0) and different treatments are indicated as: * *p* ≤ 0.05, ** *p* < 0.01, *** *p* < 0.001 and **** *p* < 0.0001.

**Table 1 cancers-13-05066-t001:** Primers used for qRT-PCR.

Genes	Forward Primers	Reverse Primers
*GAPDH*	GCAAATTCCATGGCACCGTC	AGCATCGCCCCACTTGATTT
*PTHrP*	GTCTCAGCCGCCGCCTCAA	GGAAGAATCGTCGCCGTAAA
*MIP-1α*	CTGCATCACTTGCTGCTGACA	CACTGGCTGCTCGTCTCAAAG
*ENPP2 (autotaxin)*	GGAACGCAGATGGCATGTTG	TTATCAAATCCGTGGTCTCCCT
*Wnt5a*	CTACGAGAGTGCTCGCATCC	GCCAGGTTGTACACCGTCC
*Interleukin-6 (IL-6)*	GAGGAGACTTGCCTGGTGAAA	TGGCATTTGTGGTTGGGTCA
*CXCR4*	CGGTTACCATGGAGGGGATCA	ATGACCAATCCATTGCCCACA
*TAX*	CCGCCGATCCCAAAGAAA	CCGAACATAGTCCCCCAGA
*HBZ*	AACTTACCTAGACGGCGGAC	CATGGCACAGGCAAGCATCG

## Data Availability

The data presented in this study can be made available upon request from the corresponding author.

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
