# Peer review of "The Effect of a Histone Deacetylase Inhibitor (AR-42) and Zoledronic Acid on Adult T-Cell Leukemia/Lymphoma Osteolytic Bone Tumors"

_cancers, 2021, doi:10.3390/cancers13205066_

Round 1
Reviewer 1 Report
In this manuscript, Elshafae and coworkers demonstrated that dual targeting of both ATL leukemia cells with and HDACi (AR-42) and tumor microenvironment with an osteoclast inhibitor (zoledronic acid, Zol) would decrease tumor burden in bone. They showed that AR-42, Zol, and AR-42/Zol significantly decreased the viability of a panel of ATL cancer cell lines in vitro. Moreover, Zol and AR-42/Zol decreased tumor growth in vivo, while Zol ± AR-42 significantly decreased ATL-associated bone resorption and promoted new bone formation. In contrast, AR-42 alone had no significant effect on tumor growth or osteolysis in mice. Therefore, Zol 34 adjuvant therapy might have the potential to reduce growth of ATL in bone and its associated osteolysis.
Overall, I found this manuscript well written, it presents interesting findings that support the main conclusions.
I have detected only two issues:
- In the Simple Summary, the Authors mention induction of apoptosis, but then in the Results section there is no evidence of apoptosis induction. Thus, the Authors should either amend the Simple Summary or add a figure that shows evidence of apoptosis induction.
- The Authors have investigated the effects of the drugs on the expression of mRNA levels of PTHrP, ENPP2 (autotaxin) and MIP-1a, and TAX.It would be important to support qRT-PRC results, to perform western blot analysis of at least a couple of these targets. Such an analysis would allow a better comprehesion of the changes occuring in these proteins.
Author Response
Reviewer 1:
- In the Simple Summary, the Authors mention induction of apoptosis, but then in the Results section there is no evidence of apoptosis induction. Thus, the Authors should either amend the Simple Summary or add a figure that shows evidence of apoptosis induction.
Authors’ Response:
- We really thank the reviewer for this comment, and we totally agree with the reviewer. The sentence in simple summary has been amended to be “AR-42 and Zol reduced viability of ATL cells in vitro”
- The Authors have investigated the effects of the drugs on the expression of mRNA levels of PTHrP, ENPP2 (autotaxin) and MIP-1a, and TAX. It would be important to support qRT-PRC results, to perform western blot analysis of at least a couple of these targets. Such an analysis would allow a better comprehension of the changes occurring in these proteins.
Authors’ Response:
- Although the authors agree with the reviewer that investigation of protein expression would support qRT-PCR and may elucidate further the mechanism by which these drugs work inside the cells, we think such investigation should be performed in other future studies to dissect distinct pathways affected by the drugs. Unfortunately, all of the authors have moved recently to different institutions. This would be hard to perform given these circumstances.

Reviewer 2 Report
Adult T-cell leukemia/ lymphoma (ATLL), a tumor caused by HTLV-1, is frequently associated with bone metastasis and osteolysis. In this manuscript, Elshafae et al. develop a mouse model to model osteolytic lesions observed during ATLL. For this purpose, the authors perform intratibial injections of MT-2 cells that were transduced with a luciferase gene, thus, allowing non-invasive imaging of NSG mice. Using this model, the authors analyse whether targeting both ATL cells by the use of the HDAC-inhibitor AR-42 in combination with an osteoclast inhibitor, zoledronic acid (Zol) is superior to individual treatment regimens. First, the authors test the inihibitors in HTLV-1-infected cells and find that all inhibitors affect cell vitality. After having developed their intratibial mouse model, authors show that tibias of these incoulated NSG mice show indeed marked osteolysis . Administration of AR-42, Zol, or both AR-42 + Zol revealed that AR-42 does not impact tibial tumor growth and tumor burden, while Zol and the combination AR-42+Zol significantly decreases tibial tumor growth and tumor burden. Radiography and histopathology support their findings and show that tibial bone loss and osteolysis are prevented by Zol and the comibination AR-42+Zol, but not by AR-42 treatment alone. Finally, the authors find that AR-42 and the combination of AR-42 + Zol affects transcription of the viral transactivator Tax and of cellular genes including the pro-osteolytic factor PTHrP in HTLV-1-infected MT-2 cells, which could be favourable for the tumor, while treatment with Zol alone has no impact on any of the mentioned genes.
This manuscript is interesting and tackles an important question in ATLL biology. The manuscript is well-written and the experiments are clearly described, however, there are some weaknesses in the study, e.g. only one cell line is used to study the impact of the compounds on gene expression profiles (Fig. 6) and some of the literature in the field dealing with HDAC inhibitors in HTLV-1 and ATLL biology is not cited.
Specific comments:
Major:
- 6: authors show data with one cell line. According to the journal guidelines, all experiments should be shown with at least two cell lines.
- Lines 422-430: there are many publications that have already shown that HDAC inhibitors affect expression of HTLV-1 Tax or the related BLV, therefore, it is not „unexpected“ (line 424) that AR-42 affects Tax expression. Authors should cite the relevant literature, e.g. Achachi A et al., Proc. Natl. Acad. Sci. USA 2005; Belrose G et al., Blood 2011; Schnell et al., Int J Mol Sciences 2021. Please modify this section of the discussion.
Minor:
- Line 124: MT-2 cells are not from a patient with ATL, please correct. The cells have been obtained by in vitro transformation after co-culture of CBL from a male infant with lethally irradiated ATL cells from a female ATLL patient.
- Line 126: please give more details on cell line HT1RV (Stewart et al., Virology 1996)
- Cell line ATLED is not mentioned in ref. 31, please provide correct reference.
- AR-42 is introduced as HDAC-inhibitor, author should specify that it is an pan-HDAC-inhibitor.
- Line 142: typos
- 2/3: how good was the transduction efficiency of the MT-2 cells? Is it possible that also MT-2 cells that were not transduced with the luc-gene have been inoculated? Please comment.
Author Response
Reviewer 2:
- Major:
- 6: authors show data with one cell line. According to the journal guidelines, all experiments should be shown with at least two cell lines.
Authors’ Response:
- We investigated the effect of AR42, Zol and combination on the viability of multiple cell lines (MT2, HT1RV and ATLED) in this study. We established many ATL mouse models in our previous study (Nicole A. Kohart, et al 2019, Mouse model recapitulates the phenotypic heterogeneity of human adult T-cell leukemia/lymphoma in bone”) but the MT2 cell line in the current study was superior compared to others since it 1) is superinfected with HTLV1; 2) grows well in vitro and in vivo; 3) expresses Tax; and 4) recapitulates the osteolysis in vivo in mice that is similar to ATL patients.
- Lines 422-430: there are many publications that have already shown that HDAC inhibitors affect expression of HTLV-1 Tax or the related BLV, therefore, it is not „unexpected“ (line 424) that AR-42 affects Tax expression. Authors should cite the relevant literature, e.g. Achachi A et al., Proc. Natl. Acad. Sci. USA 2005; Belrose G et al., Blood 2011; Schnell et al., Int J Mol Sciences 2021. Please modify this section of the discussion.
Authors’ Response:
- We are grateful for this valuable comment, and we totally agree with the reviewer. We modified the text and added more references to reflect the decline in TAX expression by other HDACi.
- Minor:
- Line 124: MT-2 cells are not from a patient with ATL, please correct. The cells have been obtained by in vitro transformation after co-culture of CBL from a male infant with lethally irradiated ATL cells from a female ATLL patient.
Authors’ Response:
- We totally agree with the reviewer, this sentence has been corrected.
- Line 126: please give more details on cell line HT1RV (Stewart et al., Virology 1996)
Authors’ Response:
- More details were provided on HT1RV based on the reviewer’s suggested reference.
- Cell line ATLED is not mentioned in ref. 31, please provide correct reference.
Authors’ Response:
- The reference “31” has been deleted and two new references were added instead
* Panfil, A.R., et al., PRMT5 Is Upregulated in HTLV-1-Mediated T-Cell Transformation and Selective Inhibition Alters Viral Gene Expression and Infected Cell Survival. Viruses, 2015. 8(1).
*Takeda, S., et al., Genetic and epigenetic inactivation of tax gene in adult T-cell leukemia cells. Int J Cancer, 2004. 109(4): p. 559-67.
- AR-42 is introduced as HDAC-inhibitor, author should specify that it is an pan-HDAC-inhibitor.
Authors’ Response:
- We have reflected the Pan-HDAC-inhibitor activity of AR42 throughout the text.
- Line 142: typos
Authors’ Response:
- The typos have been corrected
- 2/3: how good was the transduction efficiency of the MT-2 cells? Is it possible that also MT-2 cells that were not transduced with the luc-gene have been inoculated? Please comment.
Authors’ Response:
We are grateful for this valuable comment and understand the concern of the reviewer. We have performed many luciferase transductions in our previous studies, and we are familiar with these technical issues (Number of transduced cells and the functionality of transduction). In this study, MT2 cells transduced with luciferase vector were selected and examined before moving to our in vivo studies to maximize the number of transduced cells in the inoculum. We also performed in vitro IVIS bioluminescent imaging for the transduced cells vs parent MT2 cells to measure the efficiency of luciferase transduction and functionality and to choose the appropriate number of cells to be used in our in vivo study.
